# Re-Exploring Origins of the *Qixiang* Sacrificial Rite Practiced by the Han Army Eight Banners in Northeast China

**Lina Zhao and De Zheng \***

College of Arts, Changchun University, Satellite Road, Changchun 130022, China
* Correspondence: zhengde77@163.com

**Abstract:** *Qixiang* is a unique sacrificial culture created by the Han army eight banners in northeast China. This culture not only has elements such as shamanism and Han people burning incense, but also has military sacrificial elements. This paper argues that *Qixiang* is the evolution and legacy of *Maji*, a military sacrificial ritual in ancient China. The Han military banner people in the Qing Dynasty took *Maji* as the main body, combined the Manchu Shamanism with the Han incense burning, and created a cultural symbol representing their own ethnic group. At present, the study of *Qixiang* not only helps to understand the complexity of the development and evolution of Manchu shamanism, but also helps to reveal the ethnic identity of Han bannermen under the Eight Banners system of the Qing Dynasty.

**Keywords:** Han bannermen; *Qixiang*; rite; Manchu shamanism; *Maji*

## 1. Introduction

For a long time, scholars have mainly focused on the Manchu people and their Shaman culture in Northeast China, while ignoring a ethnic group that is still Han in nature, although it is called Manchu people. This ethnic group is the Han bannermen. They created and inherited a unique sacrificial culture—*Qixiang* (旗香), which combines a variety of cultural elements including shamanism, Han people burning incense and military sacrificial rites. Studies on this culture have been carried out since the 1980s and 1990s, and five theories have emerged.

### 1.1. Five Theories

The first theory holds that *Qixiang* originated from Manchu culture. Liu Guiteng compared the instruments, drum-beating techniques and tunes of *Qixiang* and Shamanic divine dance and found that they had many similarities. He therefore believed that *Qixiang* was a branch of Manchu Shamanism (Liu 1991, pp. 9–17).

The second theory holds that the *Qixiang* rite stemmed from the eastern expedition by Emperor Taizong of the Tang dynasty. Cheng Xun investigated the legends and folk customs in northeast China and found that the expedition to Liaodong (辽东, the area east of Liaohe River in China) during the Tang dynasty had been transformed into a local collective memory. Over time, this memory has been condensed into the story of "the king of Tang's expedition to the east" and the sacrificial ceremony of Taiping drum-beating and finally formed *Qixiang* culture (Cheng 1988, pp. 229–47). Fu Yuguang (Fu 2010, p. 175) and Yin Yushan (Yin 2016, pp. 5–8) both support this view.

The third theory is that *Qixiang* was derived from the sacrificial rite observed by the Han people. Its proponent, Ren Guangwei, discovered from his study of deity worship, ritual processes, instrumental performances and the like that the rituals of *Qixiang* were similar to those of the Nuo opera of southwest and northwest China. He thus inferred that Nuo opera evolved into the sacrificial rite of the Han people in northeast China and was absorbed by the Han bannermen to create the *Qixiang* sacrificial rite (Ren and Sun 1998, pp. 37, 51, 79, 80, 87, 89).

The fourth theory holds that *Qixiang* represents the integration of Manchu and Han cultures. Cao Lijuan studied the sacrificial objects, sacrificial environment, program structure, performers, costumes, props, music, dancing and other aspects of the *Qixiang* and found that *Qixiang* not only contains the elements of the Manchu Shamanic ritual trance dance, but also contains the elements of Han sacrifice, thus concluding that *Qixiang* is a fusion of Manchu and Han cultures (Cao 1993). Guo Shuyun also holds a similar view that Qixiang is "a form of comprehensive shamanic ritual based on the traditional ancestor worship of the Han people" (Guo 2019).

The fifth theory is that *Qixiang* was created independently by the Han bannermen. Zhang Xiaoguang analyzed the sacred prayers and ritual phrases of *Qixiang* and found that content regarding the eastern expedition during the Tang Dynasty accounted for only a small part of the rite. He argued that *Qixiang* could not have originated from the eastern expedition but was just a form of commemoration created by the Han bannermen to unite their clans and express their homesickness (Zhang 1989).

Although each of these theories can be justified, they all have their failings. In the first theory, the similarity in musical instruments cannot be seen as the reason for the Han bannermen to form a branch of Shamanic culture; in the second theory, the similarity in the deities worshipped cannot be taken as proof that the sacrificial activities associated with the Tang Dynasty Taiping drum-beating ceremony continue to be observed by descendants of the Han bannermen today; in the third theory, the similarity in rituals cannot prove that *Qixiang* originated from the sacrificial activity of burning incense among ordinary Han people; although the fourth theory integrates various streams of thought, it does not define the cultural characteristics of *Qixiang*; and the fifth theory is a bold hypothesis that highlights the Han army's talent for—and spirit of—innovation but it lacks empirical support. Thus, all of these five theories fall short in presenting a good understanding of *Qixiang* culture.

These confused definitions are detrimental to a proper understanding of the complexity of the Manchu Shamanic revival. If we regard *Qixiang* as an element of Manchu Shamanism, then we must recognize that "Manchu Shamanism revival" is not the reconstruction of Manchu and its faith in the new era, but a religious phenomenon of syncretism (Leopold and Jensen 2014, pp. 338–41), including Manchu, Mongolian, Han and other nationalities and their religious cultures. Obviously, integration is complex, and our research on *Qixiang* helps to understand this complexity. In addition, these confused definitions are not conducive to promoting the study of religious belief in Manchu society in the Qing dynasty. Although the study of the history of the Qing dynasty is grand and full of topics, the religious issue of Manchu nationality is an especially important topic for people to discuss; we have to admit that the spiritual belief of the Han army has not been thoroughly studied. If we take the Han army's acceptance of Shamanism as the only answer, it only serves to make the problem superficial and simple. In this regard, it is necessary to recognize the attributes of *Qixiang* again.

## 1.2. Military Identity and Maji ( 犸祭) Culture

The military status of the Han bannermen is key to understanding *Qixiang* culture. Although artifacts, institutions and rituals are important in cultural studies, all these elements fade into insignificance if little heed is paid to the group that participated in this kind of culture. The military status of the Han bannermen has always been neglected, and this is the main reason for continuing ignorance about the essence of *Qixiang* culture. This paper proposes that the Han army was a military group in which the Han people served and made great contributions to the founding of the Qing dynasty. The Han army was a unique and relatively independent group, whose members were neither purely Manchu nor ordinary Han. To underscore their uniqueness, they forged a distinctive military culture that included other cultural elements, such as the shamanic ritual trance dance, the incense burning of the Han people, and the *Nuoji* (傩祭) sacrificial rite, which then became *Qixiang* culture.

Thus, at its core, *Qixiang* is a military culture that evolved from the ancient *Maji* sacrificial rite as recorded in the *Book of Rites* (礼记) and the *Book of Songs* (诗经), two great ancient Chinese texts of the pre-Qin period. Wars were one of the major concerns in ancient China, which was underpinned by the belief that "the military makes a big difference to the survival of the nation" (Peng 1991, p. 211). Before marching to the battlefield, ancient Chinese would offer sacrifices to the god of war and pray for protection and victory. *Maji* sacrificial culture evolved gradually in this environment. During the Qin dynasty, the *Maji* was observed strictly as a sacrificial ceremony by the military; during the Sui and Tang dynasties, the military flag became the object of worship in the *Maji* rite (Ai 2009). This sacrificial rite survived through the Song, Yuan and Ming dynasties into the Qing dynasty. As a military group, the Han army under the Eight Banners system of the Qing dynasty not only observed the *Maji* sacrificial rite during wartime but also integrated the rite into their daily life in times of peace. Based chiefly on the *Maji* sacrificial rite, the Han army forged *Qixiang* culture, which brought together various cultural elements.

*1.3. Field Investigation and Research Methods*

This paper attempts to garner relevant information about *Qixiang* culture using three methods. The first one was field investigation. In the past few years, we have investigated Han bannermen's descendants and their sacrificial rites through observation and interviews. For instance, in February 2014, we delved into the *Qixiang* sacrificial rite practiced by the Han army Zhang clan in Wulajie Township, recording many videos and pictures, and interviewed a great number of officiants during the three-day rite. In August 2014, an investigation was made into the *Qixiang* sacrificial activities held in the Changbai Mountain Scenic Area in Jilin Province, finding the development trends of *Qixiang* culture in a modern market-oriented economic context. In January 2016, we looked into the "wake up the lake" (醒湖) ritual activity held by the Zhuanshanhu Reservoir in Jilin Province, in which the *Qixiang* officiating group led by Zhao Hongge (赵洪阁) staged a sacrificial performance; the shape of the gourd on the top of their banner effigies (旗像) made a deep impression (see Appendix A, Figure A7). In December 2020, we made another investigation into the *Qixiang* sacrificial rite by the Zhang clan in Wulajie Township, reflecting on the inheritance mechanism of *Qixiang* culture.

The second method was research on museum collections. In northeast China, many universities, counties and cities have built Shamanic culture museums, where physical collections, video clips, pictures and written materials about *Qixiang* culture are preserved, to which we paid much attention. In the process of investigation, a very valuable video of a *Qixiang* sacrificial rite recorded in 2005 was discovered.

The third method was study of local chronicles from the Republic of China. During the Republic of China (1912–1949), governments at all levels in northeast China recorded the local Han bannermen groups and *Qixiang* sacrificial rites in written text, forming what are known today as "local chronicles" of *Qixiang* culture. In this regard, a comparative study was conducted of the local chronicles, figuring out the degree to which *Qixiang* culture symbolizes the soldier status of Han bannermen.

Given this background, this paper will analyze the traces of the *Maji* sacrificial rite in *Qixiang* culture, demonstrate the connection between them and identify the basic characteristics of *Qixiang* culture.

## 2. Han Bannermen

The term "Han bannermen" refers to members of the Han army under the Eight Banners system in the Qing Dynasty. Most of them were ethnic Han. The Eight Banners in the Qing Dynasty were divided into eight army groups, namely Plain Yellow Banner, Bordered Yellow Banner, Plain White Banner, Bordered White Banner, Plain Blue Banner, Bordered Blue Banner, Plain Red Banner and Bordered Red Banner. Each group comprised Manchurians, Mongols and Hans. The army composed of Hans was called the Han Army Eight Banner. This army was in its infancy in the fifth year of the Tiancong reign of the Qing

dynasty (1631), grew in the second year of the Chongde reign (1637) and finally matured in the seventh year of the Chongde reign (1642). They were the imperial bodyguards for emperors of the Qing dynasty (Editorial Committee of Chinese History, the General Editorial Committee of the Encyclopedia of China 1992, p. 349).

The members of the Han Army Eight Banners were mostly those who voluntarily joined the later Jin (后金) dynasty at the end of the Ming dynasty or those who were captured in war by the Qing dynasty in the Liaodong region. Most of these people were Han, with a small number of Nuzhen people who had been assimilated into the Han and Mongols who were officials in the Ming Dynasty making up the rest (Yao 1995). In April of the third year of the Tianming reign (1618) of the Qing dynasty, Nurhaci, the Manchurian chieftain, attacked the Ming dynasty and launched military operations in Ningyuan, Songshan, Xingshan and Sarhu in Liaodong, capturing many Han people and inducting them into the Han army. As Hong Taiji (皇太极), Nurhaci's son, launched bigger offensives against the Ming dynasty after ascending the throne, more Han people were captured and sent to Northeast China. From the third year of the Tiancong reign (1629) of the later Jin dynasty to the seventh year of the Chongde reign (1642) of the Qing dynasty, Hong Taiji attacked the Ming dynasty five consecutive times. In 1636, 1638 and 1642 alone, he accepted the surrender of more than one million Han people. In June of the seventh year of the Chongde reign (1642), Hong Taiji created the Han Army Eight Banners, comprised of a large number of Han troops, which shared the same flag colors and official system as the Manchu Eight Banners and Mongolian Eight Banners. The Han soldiers were considered distinctly different from ordinary Han, as they were seen as conquerors who had helped consolidate the country's territorial gains (Sun 2005). Although their status was lower than that of Manchurians and Mongolians, it was much higher than that of ordinary Han people (Wu 2005).

The Han army in northeast China was divided into two types. One was called the Old Han Army, which referred to those Han who had joined the Eight Banners before the Qing armies broke through the Shanhai Pass (山海关). These people joined the Qing armies for military operations, leaving only a few of them settled in northeast China. In 1657 and 1740, a group of bannermen in the Old Han Army were transferred from Liaodong to Jilin as the Qing government set up two government agencies (the Zongguan Yamen of Dasheng Wula and the Ula Brigade Yamen) (Yin 2002, p. 42). A small number of bannermen in the Old Han Army also served in the garrison in Heilongjiang and Liaoning.

The other group was called the New Han Army and it comprised those Han who had joined the Eight Banners after the Qing armies entered the Shanhai Pass. The establishment of the Qing dynasty left northeast China a barren land. To rebuild Liaodong, the Qing government recruited people to rebuild this inhospitable land. In the first year of the Shunzhi reign (1644), the government urged farmers to participate: "the displaced people into the *Baojia* system (neighborhood administrative system), regardless of their origins, to rebuild the barren land" (Tuojin and Cao 2012). In the tenth year of the Shunzhi reign (1653), the *Regulations of Liaoning on Recruiting People for Reclamation* (辽宁招民垦荒条例) were promulgated to encourage people to move from Zhili (northern administrative region of China), Henan, Shandong, Shanxi and other provinces to northeast China. Some of these immigrants joined the Eight Banners and became the Han army (Agui and Gao 1997). For instance, according to the *Genealogy of the Lu Clan in Fengcheng* (凤城卢氏家谱), "the Lu clan in Fengcheng, with their ancestors from Lujia Dajie in Qixia County, Dengzhou Prefecture, Shandong Province, fled from Shandong Province to Northeast China because of a severe famine in the eighth year of the Shunzhi reign (1651). The Lu clan joined the Han Army of Border Yellow Banner of the Eight Banners under the jurisdiction of Shengjing" (Lu 1993).

There was another group of the Han army in northeast China, and this comprised those who were exiled for crimes during the Kangxi and Yongzheng reigns of the Qing dynasty. Volume 167 of the *Fengtian Tongzhi* (奉天通志) records that "In the early-Qing Dynasty, a total of 884 households of surrendered soldiers of Three Feudatories were

transferred from Yunnan and distributed across the borders to guard the frontiers, dig trenches, and deliver official documents to post stations"(Bai et al. 1927). The *Liaozuo Jian-wenlu* (辽左见闻录) records that "The rebellious population of Three Feudatories who had been exiled to Guandong (older name for Manchuria) came in an endless stream for several years, and they were all distributed across various posts in villages, stations, and roads" (Wang 2013, p. 173). This group of Han bannermen was also a relatively large one.

It was the Han bannermen who forged *Qixiang* culture. According to some scholars, the *Qixiang* sacrificial rite would have arisen at the time of establishment of the Han Army Eight Banners in the late Ming and early Qing dynasties (Ren and Sun 1998, pp. 37, 51, 79, 80, 87, 89). In June of the first year of the Chongde reign (1636), Hong Taiji asked officials to report immediately "if any Han people were found who claimed to be shaman and used charms and incantations to deceive people and practice witchcraft to deceive the country". Soon after,, he ordered the implementation of a unified sacrificial rite: "On June 18, following the edict of the Holy Khan, a ritual system was formulated for holding sacrificial activities in *Tangzi* (堂子, in Manchu Tangse, the Palace Temple where the private offerings of the Manchu family took place) to worship deities in a specific order" (Erdeni and Kuerlen 1990, pp. 1512, 1514). In addition to the officially prescribed sacrificial rituals, "other forms of arbitrary sacrifice are prohibited forever". This prohibition confirms the fact that burning incense was a common practice among the Han people (including the Han bannermen). When the Manchurians entered the Shanhai Pass, the activities of burning incense and offering sacrifices gained even more momentum among the Han bannermen. The *Altar Records* (坛续), written by the Zhang clan, now available in Wulajie Township, Jilin Province, record the emergence of the *Qixiang* rite during the Kangxi reign in the Qing dynasty: "During the reign of Emperor Kangxi of the Qing Dynasty, there was a Liaoyang man named Yang Zhong. He was smart and studious since childhood. In his youth, he met a talented person who taught him exceptional martial skills and breathing techniques. From then on, he practiced day and night according to the motion laws of star clusters. He endured the hardships of an arduous journey and devoted himself to practice in the Kanli Cave of Changbai Mountain. After two decades, he returned to society and assumed the Taoist name of Old Master Qingyang. He taught people how to set up altars to worship gods and how to beat drums rhythmically to invite the gods to religious rituals. He imparted his profound knowledge to disciples from eight clans: Wang, Liao, Chang, Zeng, Xie, Wan, Hu and Hou, and this gave rise to the Eight Altars of Guandong". With the passage of time, the Eight Altars saw countless successors. This sacrificial rite of the Han army spread across Guandong. This shows that holding the *Qixiang* rite was a common practice in the everyday lives of the Han bannermen of the Qing dynasty.

### 3. *Qixiang* Sacrificial Rite

At present, only a few groups are familiar with the routines of a complete *Qixiang* sacrificial rite, and most of them are found in rural areas, such as Wulajie Township and Yongji County of Jilin Province in northeast China. A complete sacrificial ceremony takes three days and involves four stages.

Stage 1: Offering livestock sacrifices and putting together all items necessary for the rite.

This part of the rite, carried out during the day and at twilight of the first day, mainly involves preparations and includes the arranging of artifacts, painting new portraits, offering livestock to the deities and making banner effigies (旗像). The artifacts include a bow, three arrows, two hay cutters, a pig head, three stacks of steamed bread, two "holy bottles", snacks, fruits and bouquets of yellow incense (see Appendix A, Figure A2). "Painting new portraits" means that the household owner burns the old portraits of deities and then invites someone to paint new ones. "Offering livestock" refers to the sacrifice made with black pigs by the head of the household to ancestors and deities. They pour wine into the pig's ear before killing the pig, and the pig will shake its head and ears, indicating that the ancestors and deities have received the pious wishes of the householder. They then kill the pig and sacrifice it with all the meat (see Appendix A, Figure A1). When the divine

craftsman (神匠) (who presides over the ceremony) invited by the householder arrives, they make the banner effigies with colored paper, corn straw and other materials.

The banner effigy is a unique implement of the *Qixiang* rite, which is not seen in any other kind of sacrificial activities. It symbolizes the flag, camp, generals, money, grain and military orders, etc. of gods and ghosts. When it is made, it means that a sacred space where people and gods co-exist has been constructed in the courtyard of the head of the household. In general sacrificial activities, two banner effigies (one high and one low) should be made, the high one is called the "god flag image" and the short one is called the "ghost flag image", while in special sacrificial activities where the master enrolls disciples, four flag images (two high and two low) should be made, two of which belong to the master and the other two belonging to the disciple, which means that the master grants the power to command people and gods to the disciple. In 2014, we saw four banner effigies in the sacrificial rite performed by the Zhang clan (张氏家族, the former Old Chen Hanjun Bannerman) in Gongtong Village, Wulajie Township, Jilin Province. Each effigy was composed of a flag top, collar, coned garment (made of colored paper) and pole. The flag top was made of hardboard and shaped like a pointed spearhead (Figure 1). The shape, structure and symbol used in the banner effigies bore a striking resemblance to the "*Dadao*" (大纛 The most authoritative flag in the army) used in ancient times. In this ceremony, the Zhang clan's officiating master awarded the "*Dadao*" to the new generation of inheritors, giving them the right to take charge of the clan's sacrificial rite.

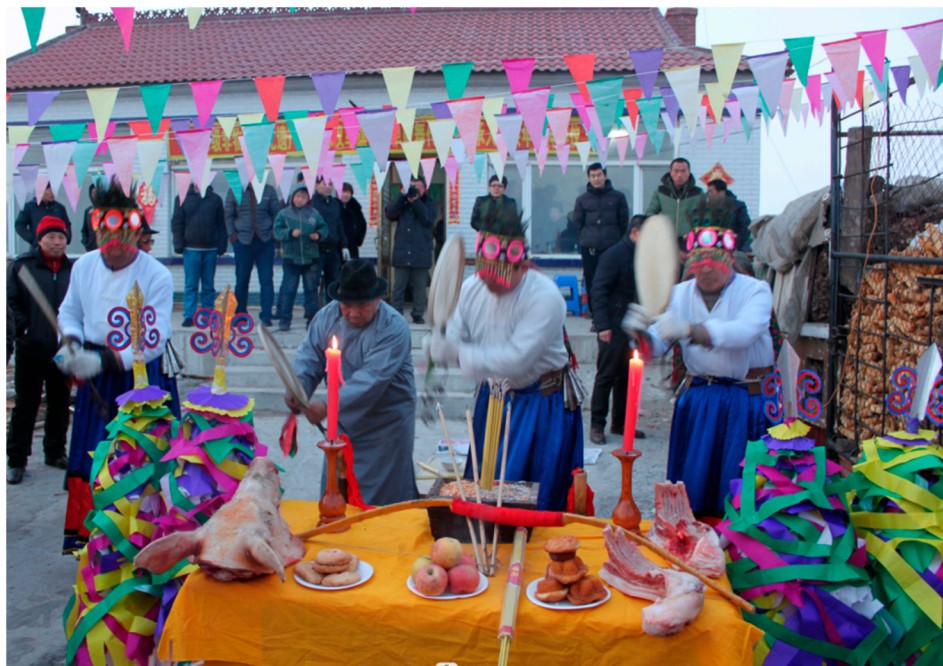

**Figure 1.** Four banner effigies placed on either side of the table with offerings at the Zhang clan's *Qixiang* sacrificial rite. Photograph by Zheng De on 17 February 2014.

Ren Guangwei observed and recorded another form of banner effigy in the *Qixiang* sacrificial rite of the Leng clan in Xinbin County, Liaoning Province. This kind of banner effigy was directly called the "*Dadao*" by the local Han bannermen, which is consistent with its name in the ancient army. Its shape is as follows:

As the iconic feature of the *Qixiang* rite, the *Dadao* comes with a frame made of sorghum stalks. A grasshopper cage-shaped frame is first woven with sorghum stalks, with four thick stalks inserted obliquely into the cage as four legs to make it stand firmly on the ground. Then, multi-colored paper is cut into ripple-shaped fishing nets (the ripples are supposed to represent the "rising tide of seawater" and allude to the Emperor Taizong of the Tang dynasty conquering northeast China via the sea route) and placed layer by layer

on the framework. The framework with the nets on it, measuring 1.5 m high, is called the "camp" for the soldiers of the Tang Dynasty. On the top of the "camp" are two lotus flowers crafted with colored paper, which are overlapped to represent the memorial ceremony. On the flowers sits an eight-sided *diaodou* (刁斗, copper army pot) made of gold and silver foil, which serves as an altar. Inside the *diaodou*, standing upright, is a triangular wolf-teeth flag made of yellow paper or cloth, on which is a circle with the Chinese character "tang" (representing Li Shimin 李世民, Emperor Taizong of the Tang dynasty) or "wang" (representing Wang Junke 王君可, a general of the Tang Dynasty) in its center. Red, blue, white and black pennants are inserted in the four directions, which, along with the yellow flag, represent the Chinese Five Elements as well as the Tang soldiers coming from five directions. On the top of the "camp" sits the big army banner of the Tang camp, which stands 50 cm high. The total height of the flag is 2 to 2.5 m. There are two *Dadao* in this form. In most cases, they are placed on either side of the front gate of the host family, while sometimes they stand on the two sides of the altar in the courtyard. They are burnt after the completion of the incense-burning ceremony (Ren and Sun 1998, pp. 37, 51, 79, 80, 87, 89).

In contrast, the banner effigies in Xinbin Town, Liaoning Province, have greater military significance than those in Wurajie Town, Jilin Province, and are more similar to the ancient military flag.

Stage 2: Inviting the gods.

This stage of the rite extends from the night of the first day through the morning of the next day and involves the specific activities of *Jieshenxiang* (接神像, welcoming divine portraits), *Songjian* (送箭, shooting arrows), *Anzuo* (安座, placing the portraits of gods in a specific order), *Nianshen* (念神, chanting prayers) and *Dawulu* (打五路, opening the five paths).

"*Jieshenxiang*" refers to inviting the ghosts of ancestors and gods back home in the form of portraits that are worshipped (see Appendix A, Figure A4). When investigating the Qixiang ceremony of the Zhang family, the author found a total of 15 portraits, including: *Xiangfeng* (先锋), *Guangye* (关爷), *Shangshen* (上神), *Wangzi* (王子), *Qishen* (旗神), *Hushen* (虎神 tiger god), *Fengdu* (丰都 the god of the netherworld), *Jiatang* (家堂 the genealogy with the names of ancestors), *Yanguang* (眼光 the goddess of eye disease control), *Wudao* (五道 the god that drives away ghosts), *Choujin* (抽筋 the god that controls muscle cramps), *Chouchang* (抽肠 the god that controls abdominal pain), *Kelao* (咳唠 the god that controls coughs), *Toutong* (头痛 the god that controls headaches), *and Erming* (耳鸣 the god that controls tinnitus).

Of these, the first five are associated with military campaigns. *Xianfeng* refers to Xue Rengui (薛仁贵), a celebrated general of the Tang dynasty (618–907); *Guanye* to Guan Yu, a famous general in the Three Kingdoms Period (220–280); "*Wangzi*" to Li Shimin, the emperor of the Tang dynasty; *Shangshen* to the 120 generals of Li Shimin's military expedition to Liaodong and *Qishen* to the god of the *Dadao* worshipped by the Ming armies in the *Maji* sacrificial rite. These five deities are placed in front of the shrine and worshiped first, demonstrating their very important position. Other gods, such as *Hushen* and *Taiwei* originated from Shamanism and are worshipped to appease wild animals; *Fengdu*, *Yanguang* and *Wudao* are the gods of Taoism and of folk religions and *Kelao*, *Erming*, *Chouchang*, *Toutong* and *Choujin* are the gods who control diseases of body organs. These gods rank behind the military gods, indicating their slightly lower status. "*Jiatang*" (Figure 2, covered with red cloth) is the ancestor god of the household head's family and is ranked higher than the gods responsible for health but lower the military gods. (Figure 2, arranged from right to left).

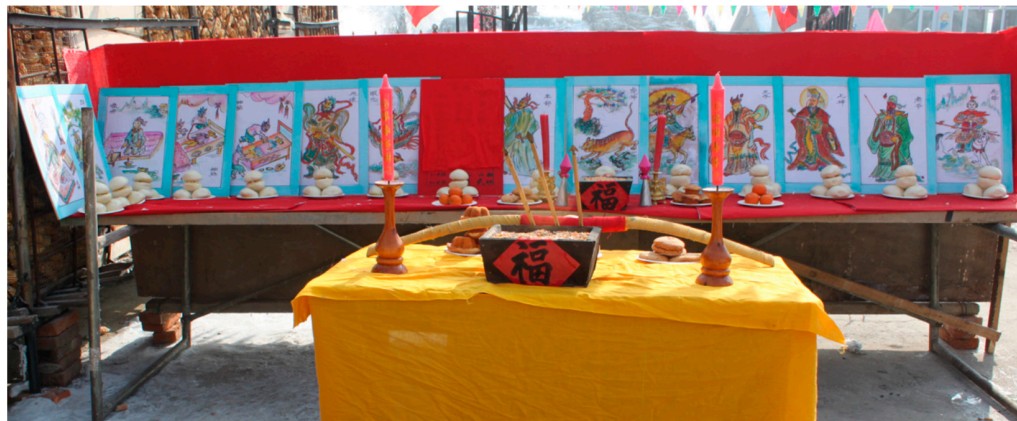

**Figure 2.** In the Zhang clan's *Qixiang* sacrificial rite, 15 portraits were placed on the god table, with Jiatang (covered by red cloth) in the middle. From right to left, they are *Xiangfeng, Guangye, Shangshen, Wangzi, Qishen, Hushen, Fengdu, Jiatang, Yanguang, Wudao, Choujin, Chouchang, Kelao, Toutong, and Erming.* Photograph by Zheng De on 17 February 2014.

"*Songjian*" means that the householder shot three arrows at the lintel of the main hall of the house. This part of the rite alludes to the legendary story of General Xue Rengui's "three arrows as a mark of resolution to defend the frontier" in the Tang dynasty. The symbolic meaning is to show that the leader of the gods, *Xianfeng*, demonstrated his martial arts to calm the war in the Tianshan Mountains, shocking all the gods and dead souls who have been invited into the house so that they cannot move at will, and ensuring the safety of the head of household and his family. "*Anzuo*" refers to the placing of the portraits of the deities in the shrine in the correct order. "*Nianshen*" means that the person leading the rite will chant the words of the gods in the form of divine songs for the host family (see Appendix A, Figure A5). "*Dawulu*" refers to the officiant performing the martial skills of the legendary general Wang Junke using a broadsword. In essence, the rituals and gods observed and worshiped in the second stage have more military elements.

Stage 3: *Fangshen* (放神, performing the role of gods).

This stage lasts throughout the second day until midnight and covers the rituals of *Qijian* (启箭), *Kaijinkou* (开金口), *Fangshen* and *Shaozhi* (burning paper money). "*Qijian*" refers to the taking down in the morning of the three arrows that were shot the previous night into the main hall of the house to lift *Xianfeng*'s order, indicating that the other gods and dead souls can move at will. "*Kaijinkou*" refers to the officiant worshiping the ancestors with chicken blood in front of the shrine and decorating the ancestors with combs and colorful flowers.

"*Fangshen*" means that the officiant performs the spirit appendage, displaying the magic and martial skills of *Xianfeng, Wangzi, Taiwei, Yingsheng* (鹰神, eagle god), *Fengdu, Hushen, Jinhuahuoshen* (金花火神, the god of fire) and *Wudao*. The officiant first performs the role of *Xianfeng*—he walks to a horse, feeds it, saddles it, tightens the girth, puts on the bridle, pulls the reins, mounts the horse, and shoots arrows in a symbolic demonstration of the martial skills of marching and fighting. The second role to play is that of *Wangzi*. The officiant mimics the actions and facial expressions of Li Shimin, the Tang dynasty emperor, when he saw soldiers die and beat his breast and stamped his feet in sorrow. Other performances are related to deities linked to animals and Taoist or folk deities. Thus, the first performances that symbolize the roles of *Xianfeng* and *Wangz*i were of great importance because they refer to military operations.

At midnight, the officiant leads the head of household and his family in burning the banner effigies and throwing the sacrificial food and wine into the fire, indicating that money and food have been delivered to the gods and late relatives. This is what is referred to as "*Shaozhiqian*." In this ritual, special attention is paid to the role of the banner effigies. The erection of the banner effigies indicates the creation of a sacred space for the

coexistence of gods and humans in the courtyard. When these effigies are burned, the sacred space also goes with them. In this sense, the banner effigies play a pivotal role in the *Qixiang* sacrificial rite (see Appendix A, Figure A6).

Stage 4: Warnings and taboos.

This stage lasts from the early morning of the third day until noon. After daybreak, the head of the household hangs the *Xiliu* (喜绺, a straw pole tied with red rope) on the door, indicating that the family held incense-burning and sacrificial activities. Black cattle, white horses, people in mourning, and unclean people were forbidden to enter, hoping that passers-by would know. A month later, the householder takes down the *Xiliu* and throws it into a clean, isolated place. This signals the wrap-up of the sacrificial rite.

In the above four stages of the *Qixiang* sacrificial rite, we found that the most important symbolic objects were banner effigies (see Appendix A, Figure A3), while *Xianfeng*, *Wangzi*, *Qishen*, *Shangshen* and *Guanye* were the most-worshipped gods and "*Songjian*" and "*Qijian*" were the most military-order sacrificial ceremonies. All performances imitating gods are full of military flaunting elements. These characteristics show that the *Qixiang* is closely related to the ancient *Maji* sacrificial rite.

## 4. *Maji* Elements in the *Qixiang* Sacrificial Rite

### 4.1. Maji Was a Military Sacrificial Ceremony Performed in Ancient China

Firstly, *Maji* was a sacrificial rite that was held in ancient China before the launch of a military operation. The *Songshi* (宋史) says, "*Maji* is a kind of military sacrificial rite that overrides all other types of military rituals" (Tuotuo and Alutu 1985, p. 2829). *Maji* was highly valued by the ancients, and livestock and even people were used in the sacrifice. The *Yuanshi-liezhuan* (元史●列传) recorded that "On June 30, … Dali (答里) responded with war, killing the messenger HarHarHarun Aruhui to sacrifice the military flag" (Song 2001, p. 3334).

Secondly, from the Sui Dynasty (581–618) and the Tang Dynasty (618–907), the state took the military flag as the object of sacrifice and formed a complete system. In the etiquette system of the Song Dynasty, the flag that led the army forward was called *Ya* (牙A large flag with a zigzag shape at the edge), and the army must hold a sacrifice to it before the expedition. This system reached its peak in the Ming Dynasty (1368–1644). In order to worship the military flag, the Ming government set up flag temples across the country, dedicated to the worship of *Dadao* in the army and called *Dadao* "Flag Head General" (Guo 2013).

Thirdly, there were music and dance performances in the *Maji* sacrificial rite. According to the *Xihu Liulan Zhi* (西湖浏览志) of the Ming dynasty, "At the flag temple … on the day before the military flag sacrifice ceremony, the soldiers marched around the city with weapons to display, and the drums and orchestras were played frequently, which was called "show the army". On this day, people show various skills and compete with each other, which is very lively (Tian 1991, p. 229). Obviously, the *Maji* rite had developed into a folk custom festival dominated by military elements, promoted by state officials and participated in by ordinary people in the Ming dynasty.

In military activities or wars, in order to achieve the desire of victory, the ancient Chinese placed their hopes on the flag god, which prompted them to worship the flag god regardless of the changes of the dynasty. In this regard, *Maji* rite has strong vitality.

### 4.2. Maji Was Preserved by the Han Bannermen

Since the Han bannermen of the Qing dynasty were all soldiers, they inevitably believed in *Maji* culture. On the one hand, they inherited the traditional practice of the *Maji* sacrificial rite from the Ming dynasty. In the battle with the Qing army, they failed and surrendered. Although the goal of the battle has changed, as soldiers, they still maintain the belief of sacrificing the flag. Therefore, the Han troops who performed the *Maji* sacrificial rite during the Ming Dynasty retained their tradition of worshiping the flag after their surrender to the Qing Dynasty.

On the other hand, the Han bannermen had to follow the system of the *Maji* sacrificial rite in the Qing dynasty. Volume 84 of the *Qing Shi Gao* (清史稿) records: "To celebrate the pacification of Shenyang in the tenth year of the Tianming reign, the Manchu armies withdrew to the Huhun River to kill cattle and worship the banner. The armies waged a war against Korea in the first year of the Tiancong reign (1627) and returned triumphantly in the second year, with a big banner erected to worship the heavens. Since then, the armies have often worshiped the banner for a military operation or battle victory, and the banner worshipped every time is enshrined in the Guandi Temple" (Wang 1977). This shows that the *Maji* rite in the Qing dynasty had also become a state system, which lasted until the end of the dynasty. As soldiers of the Qing dynasty, the Han bannermen had to follow this system. The *Suijun Jixing* (随军纪行) says: "On August 29, the general led all *jalan i janggins* (regiment captains) and *janggins* (captains), among others, to kill eight oxen and offer them on the earthen altar while blowing the conch shell horns in worship of the banner" (Zeng 1987, p. 5). This is a historical record of how the Han bannermen worshiped the big army banner, demonstrating that the *Maji* rite was quite a common practice at that time.

For Han bannermen, *Maji* sacrifice is not only the military system of the country but also the spiritual belief of the ethnic group. It entered the field of life from the military field and has become the identity symbol of Han bannermen—*Qixiang*. Stephan Feuchtwang believes that folk religion is the metaphor of empire. People imitate the orthodox politics of the empire by means of metaphor, complete the construction of self-worth and ethnic identity in the form of folk religion (such as festivals, temples, genealogy, ancestral halls, worship, etc.) and create a new meaning to redefine imperial power (Feuchtwang 2001, pp. 71–104). Although this theory does not answer the question of the legitimacy of imperial authority, it relatively accurately describes the relationship between folk religion and national politics. Looking at *Maji* and *Qixiang* from this perspective, we can explain the relationship between the military gods, martial arts performances (such as throwing spears, shooting arrows and wielding the sword, etc.), banner effigies and *Maji* rituals.

### 4.3. The Dadao in Maji Evolved into the Banner Effigies in Qixiang

The banner effigies are the most important object in *Qixiang* and have many similarities with *Dadao*.

Firstly, both share almost the same nomenclature. When Mr. Ren Guangwei studied *Qixiang* in Liaoning Province, he found that the local people called the banner effigies "*Dadao*" (Ren and Sun 1998, pp. 37, 51, 79, 80, 87, 89), which is consistent with the name of *Dadao*, the object of *Maji* sacrifice in the ancient army. Zhang and Chang, the Han bannermen in Jilin Province, called the banner effigies "flag logans" and "benchmarks", to refer to the icons of the general and his commands for operations, which was the basic function of *Dadao* in the ancient army. In addition, and more importantly, in the Zhang's god system, there is a god called "flag god", which is one of the most important gods (Zheng 2021, p. 254), and its source should be the "Flag Head General" god in the Ming Dynasty *Maji* rite.

Secondly, both share a similar structure. According to ancient Chinese military books, *Dadao* was usually decorated with gourd-shaped or pointed spearhead parts. The *Jixiao Xinshu* (纪效新书) of the Ming dynasty records that "the Banner of the Middle Troop" was decorated with a gourd-shaped head (Figure 3), and "the Flag of the Commanding General" was decorated with a pointed spearhead (Figure 4). The "Banner of the New Gun Camp of Plain Yellow Banner", as seen in the *Qinding Daqing Huidiantu* (钦定大清会典), was decorated with a gourd-shaped head (Figure 5). This decorative element also exists in the banner effigies of Han bannermen. The top of the banner effigies of the Chang clan in Yongji County, Jilin Province, is shaped like a gourd, while the banner effigies of the Zhang clan in Wulajie Township is shaped like a pointed spearhead, which is consistent with the *Dadao* recorded in ancient books. In addition, the *Taibai Yin Classic* (太白阴经) of Li Quan of the Tang Dynasty recorded that "there are five flags in five directions, each with its own color . . . They are placed behind the six big banners in a military march or in the military camp" (Li 2007). The so-called "Five Direction Flags" refer to the five colors

of blue, red, white, black and yellow, representing the five directions of east, south, west, north, and center, respectively. The meaning of the "Five-faced Tooth Banner" of the Leng clan's *Dadao* in Liaoning Province is aligned with that of the "Five Direction Flags" in this book.

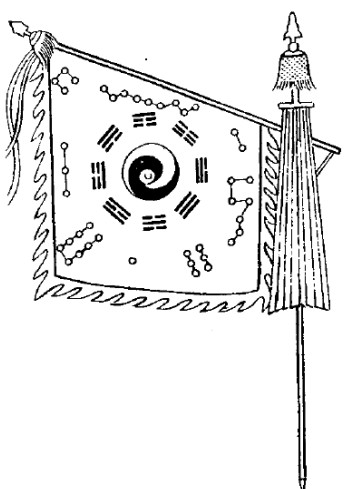

**Figure 3.** Banner of the Middle Troop (Qi 2017).

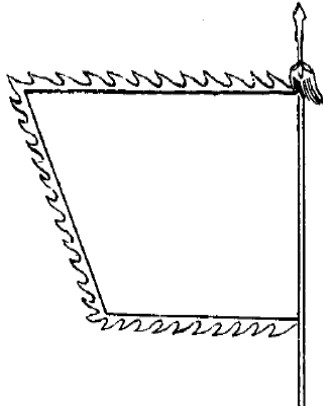

**Figure 4.** Flag of the Commanding General (Qi 2017).

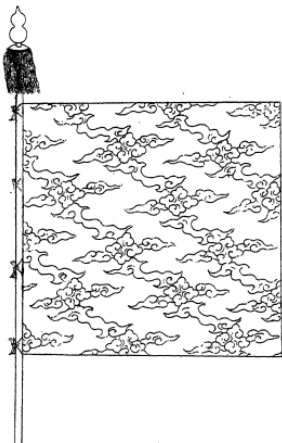

**Figure 5.** Banner of the New Gun Camp of Plain Yellow Banner (Kungang and Li 1899).

Thirdly, both share similar functions. The *Dadao* is located at the place where soldiers gather and where the command post of the general stands. Likewise, in the *Qixiang* sac-

rificial rite, the sacred space created by the banner effigies brings together not only those living but also the souls of dead soldiers and generals. In this respect, the two are completely consistent.

Fourthly, both share a similar symbolic significance. The *Dadao* is not only an object of adoration by the soldiers, but also symbolizes the military generals' power to authorize troops to wage war. Similarly, in the *Qixiang* sacrificial rite, the banner effigies symbolize the master's power to authorize disciples to command the people and gods. The two are consistent in this respect.

### 4.4. The Space Dedicated to the Maji Rite Evolved into the Space for the Qixiang Rite

*Maji* was a national military sacrificial ceremony, which was held in military camps, training grounds or flag temples (Guo 2013). These places evolved and condensed into the family courtyard marked by the flag image in the *Qixiang* ceremony.

Firstly, the place where the banner effigies are located symbolizes the military camp. The Leng Divine Song of Liaoning Province says: "A pair of *Dadao* stand squarely in front of the gate. Flags on the poles with the *diaodou* (刁斗, copper army pot) on it are flying in the wind. Wang Junke has issued a military order to mobilize his soldiers under the banner. The general has given the command to pitch a camp under the *Dadao*, with the ghosts of soldiers prohibited from breaking out of the camp" (Ren and Sun 1998, pp. 37, 51, 79, 80, 87, 89). Judging from this song, the banner effigies set up at the gate of the courtyard are like the erecting of the *Dadao*, showing that a military camp has been established.

Secondly, the place where the banner effigies are located symbolizes the *Yamen*. (衙门, government office). At the beginning of the *Qixiang* sacrificial rite, the Zhang clan in Wulajie Township, Jilin, sang a divine song that went like this: "New gods, horses, and banners are invited to the new altar, and new gods and horses are welcomed to settle in your *Yamen*." The reason why they use "*Yamen*" here to refer to the ordinary household courtyard is that the Han bannermen deemed the place where there is a "new flag" (namely the banner effigy) to be a *Yamen* that could command large groups of troops and horses, instead of being just a place for ordinary life.

Thirdly, the place where the banner effigies are located symbolizes the temple. The Leng Divine Song of Liaoning Province states: "My emperor, my lord, please quickly gather the ghosts and order them to settle in the temple. The Emperor of Tang Dynasty has signed a decree to announce outside the tent that 3000 souls of loyal soldiers should hear clearly that you are admitted to any temple you see in *Guandong* (关东, refers to the area east of Shanhai Pass in China) and can be worshipped only at those temples" (Ren and Sun 1998, pp. 37, 51, 79, 80, 87, 89). According to the song lyrics, when the divine craftsman finished making and erecting the banner effigies, it meant that the courtyard was converted into a temple, and all kinds of gods and dead souls would enter this sacred space.

### 4.5. The Flag God in Maji Evolved into the Qishen (Flag God) in Qixiang

The gods of the flag in the *Maji* of the Ming Dynasty include "Flag Head General", "Six Flag General", and "Five Direction Flag" (Guo 2013). After the fall of the Ming dynasty, these gods were continued and integrated into one in the *Qixiang* sacrifice of Han bannermen, which is called "*Qinshen*". As mentioned earlier, *Qinshen* has a high position among all the deities worshipped, and is one of the few major gods to be worshipped in the form of divine portraits, showing a unique significance.

### 4.6. The Drum-Beating in Maji Evolved into the Single-Drum-Beating Dance in Qixiang

As with the *Maji* sacrificial rite, drum-beating, juggling, weapons display and martial arts performances also featured in the *Qixiang* ceremony. The officiant not only performed a single-drum-beating dance and shook waist bells but also presented dramatic performances to show the activities of the gods. In addition, the officiant showed off various weapons such as bows and arrows, broadswords, and spears, as well as displaying such

arts as archery, sword fights and breaking pigs' heads. All the rituals were accompanied by music and chants, creating a scene very similar to that of the *Maji* rite.

## 5. *Qixiang* Evolved from *Maji*

Many details show that *Qixiang* sacrifice originated from the ancient *Maji*, which can be described and analyzed as follows:

> After the founding of the Qing dynasty, the Han bannermen in northeast China became a group with unique emotional needs and were different from both the Manchu and ordinary Han people. They were the people who helped the Manchus win the war, so they declared their war achievements to the rulers and hoped to be respected, but they were also the people who submitted to the Manchu army, so they must express their obedience to the supreme ruler to gain trust. Their political and economic status was higher than that of ordinary Han people, so they show their sense of superiority in various ways, but they still belong to the Han nationality, so they must rely on Han culture for spiritual and emotional support. These attributes determine that Han bannermen must take *Maji* as the core, integrate Manchu Shaman dance and Han people's incense-burning, and create the cultural symbol of their own ethnic group—*Qixiang*.

The Han bannermen imitated and reshaped a new *Maji* rite in folk life. From the perspective of "Empire Metaphor" (Feuchtwang 2001, pp. 71–104), *Maji* is one of the orthodox military politics of the empire, and the Han bannermen are both participants and imitators of this politics. The so-called participants refer to Han bannermen's participation in orthodox sacrificial activities as national soldiers and the so-called imitators refer to the Han bannermen's integration of sacrificial rites into life as a folk mass. This dual identity is not the intention of the Han bannermen, but was determined by the social and political environment, that is, the nature of military and civilian integration of the "Eight Banners" system (八旗制度) in the Qing dynasty. Therefore, the Han bannermen were both soldiers and civilians, and their religious symbols needed to adopt the *Maji* sacrificial element.

At the same time, the attribute of subordinating to the Manchu rulers determined that Han bannermen must accept Manchu Shamanist beliefs while the emotional dependence of Han culture required that they must accept the Han people's incense-burning sacrifice. These two and *Maji* were integrated into each other to form the *Qixiang* sacrifice. This phenomenon can be explained by the so-called religious syncretism theory of Leopold and other scholars, which believes that the cultural symbols shared by different human groups are often fused together in another way, showing different meanings, so "it is no great leap to declare that all religion is syncretic" (Leopold and Jensen 2014, pp. 338–41; Light 2014). From this theory, *Qixiang* is indeed a product of the syncretism of multiple religious cultures.

However, we must see that much important information contained in the *Qixiang* still has not been effectively interpreted. For example, what kind of emotion does the *Qixiang* ceremony, as the cultural symbol of the Han bannermen, express to the ruler? Does the *Qixiang* mark that the Han bannermen have become an independent ethnic group? What kind of Chinese-style religious belief system does the *Qixiang* ceremony contain? Is it only because the ruler is the winner that the Han bannermen imitate the *Maji* sacrifice? Does the evolution and formation of *Qixiang* indicate the direction of the evolution of Shamanism in China? These issues are obviously very important and are closely related to China's political, historical and cultural traditions, especially the history of the Qing dynasty, Shamanism and folk beliefs. In order to answer these questions, it is necessary to learn from Qu Feng's viewpoint of reconstructing the Shamanism theory through a dialogue between China and the West (Qu 2018), based on the connotation of China's "civilization" and around the theme of *Qixiang* culture, to carry out "dialogue" in the fields of modernity and tradition, reality and history, politics and religion, and to obtain practical answers.

**Author Contributions:** L.Z.: Funding acquisition, Investigation, Data curation, Resources, Writing, original draft. D.Z.: Conceptualization, Methodology, Writing, review & editing. All authors have read and agreed to the published version of the manuscript.

**Funding:** This research received no external funding.

**Institutional Review Board Statement:** Not applicable.

**Informed Consent Statement:** Not applicable.

**Data Availability Statement:** Not applicable.

**Acknowledgments:** I am grateful to reviewers for their constructive comments. Thanks are also given to Thomas Michael and Feng Qu for their editing work.

**Conflicts of Interest:** The authors declare no conflict of interest.

## Appendix A. Field Research Picture

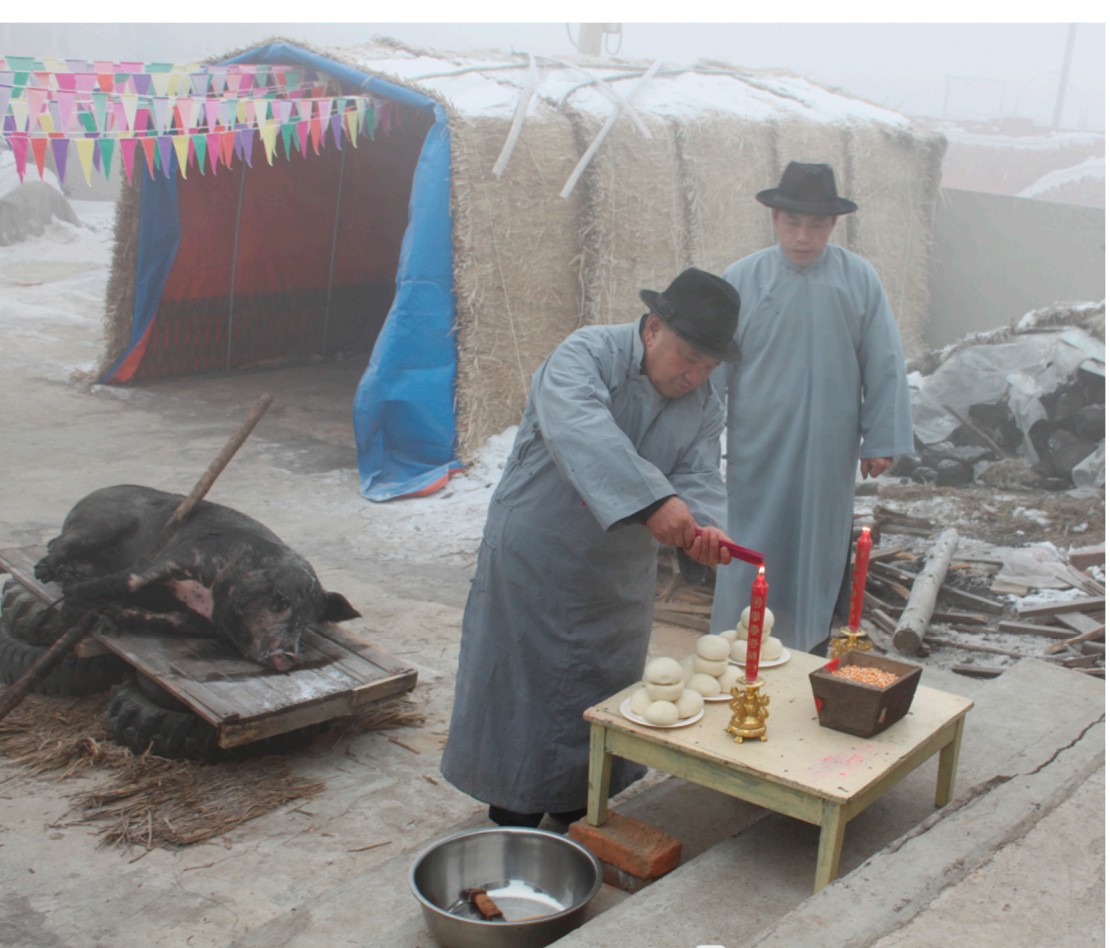

**Figure A1.** In the preparation stage, Zhang Zonghua (张宗华), the master of the altar, sacrificed a black pig to the gods and ancestors (17 February 2014, *Qixiang* sacrificial activity of the Zhang family in Wulajie Town, Jilin Province). Photo: De Zheng, 2014.

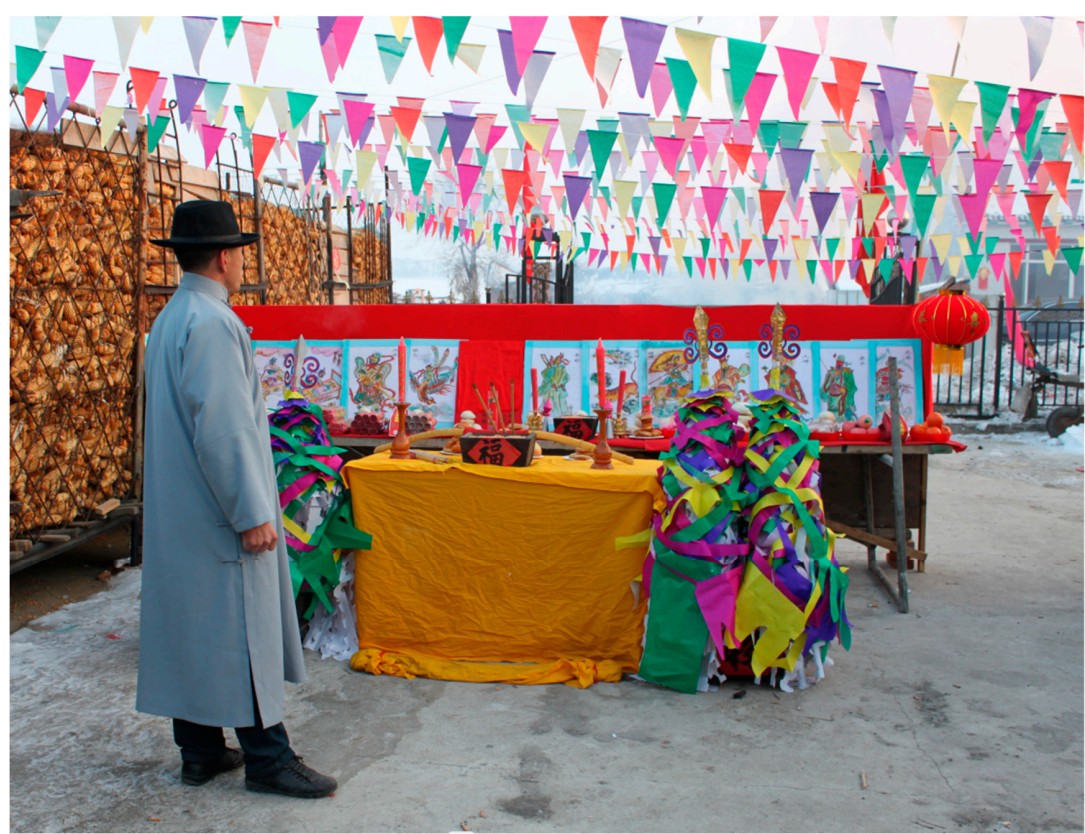

**Figure A2.** In the courtyard, four banner effigies, divine portraits and offerings have been prepared. On the table lie a bow and three arrows (17 February 2014, *Qixiang* sacrificial activity of the Zhang family in Wulajie Town, Jilin Province). Photo: Zheng De, 2014.

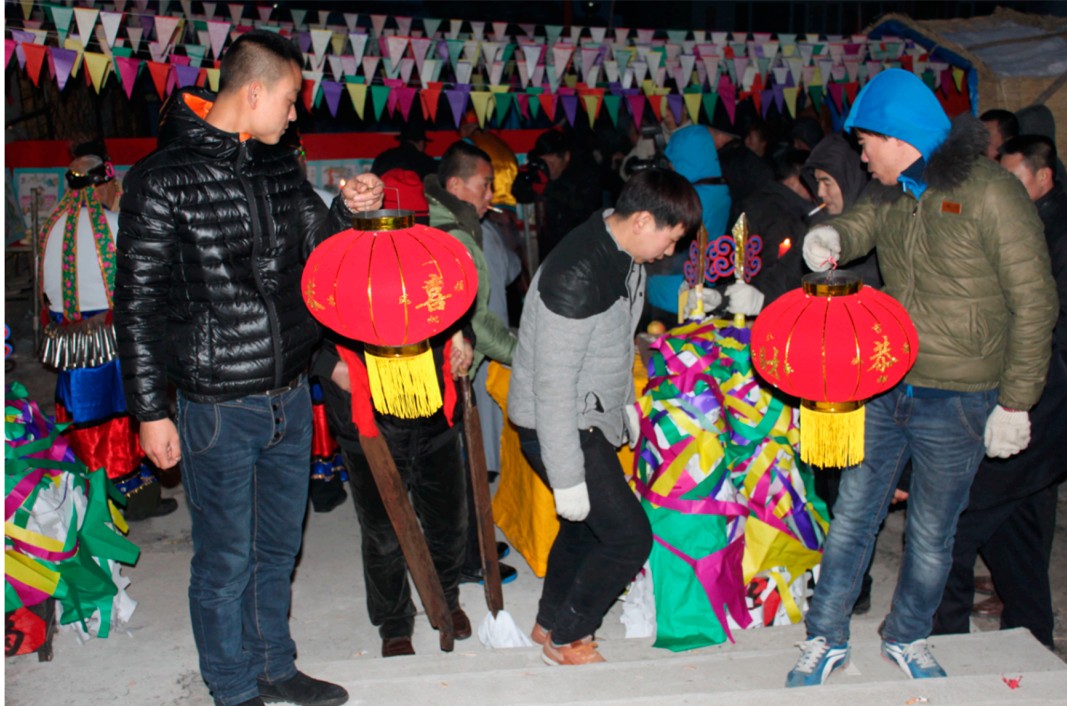

**Figure A3.** People took the banner effigies from the yard to the house, symbolizing that under the leadership of *Dadao*, gods and ghosts were invited to the house (17 February 2014, *Qixiang* sacrificial activity of the Zhang family in Wulajie Town, Jilin Province). Photo: Zheng De, 2014.

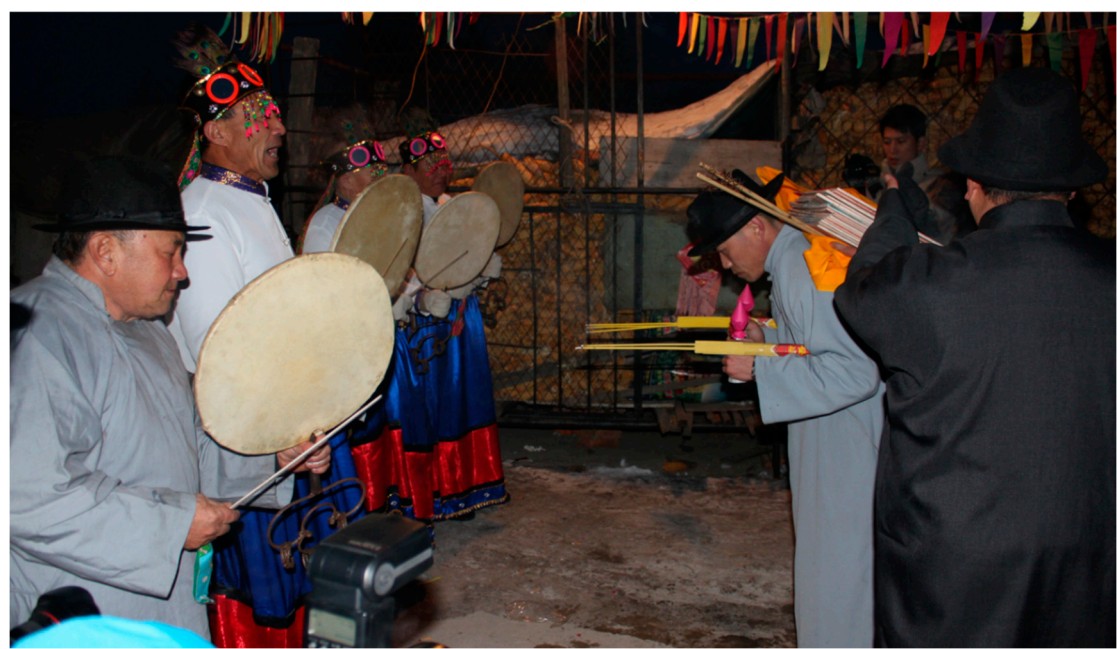

**Figure A4.** Zhang Hongnian (张宏年), the son of Zhang Zonghua, carrying a bow and arrow across his body, holding yellow incense under his arms and a treasure bottle in his hands, bending down and carrying all the divine portraits (including military gods, tiger gods, wild boar gods, etc.) and ready to follow the banner effigies into the house. His gestures and actions expressed his piety and awe for gods and ancestors (17 February 2014, *Qixiang* sacrificial activity of the Zhang family in Wulajie Town, Jilin Province). Photo: Zheng De, 2014.

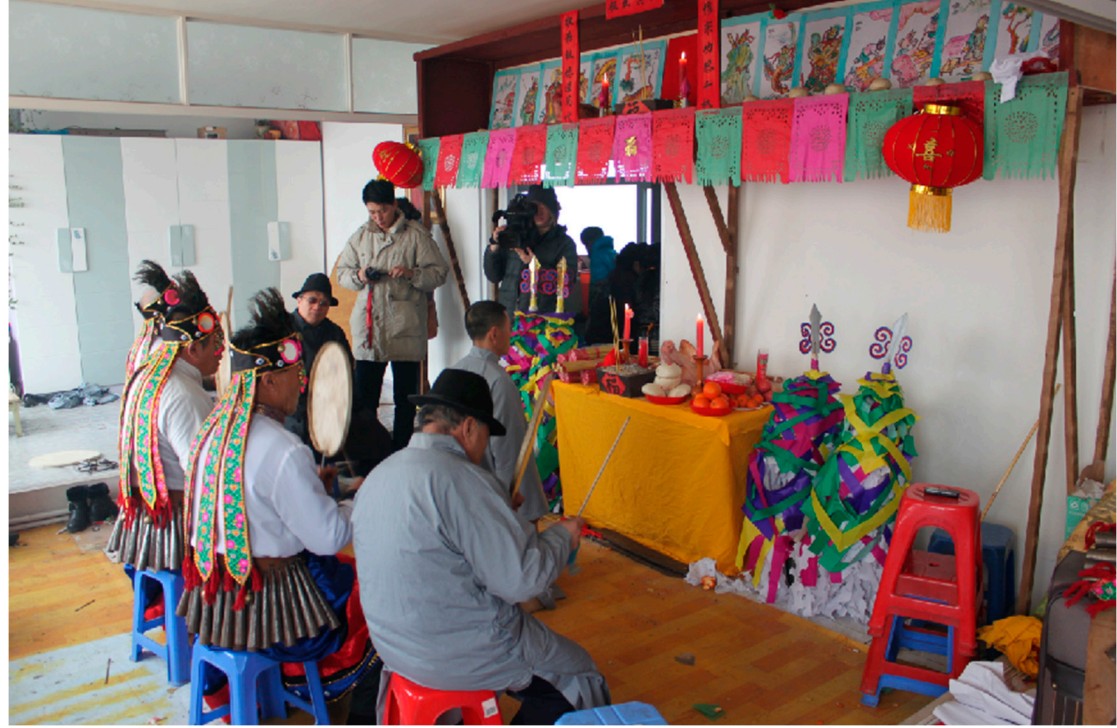

**Figure A5.** The divine portraits were consecrated in the shrine. The craftsmen beat drums and sang divine songs in front of the shrine and banner effigies (18 February 2014, *Qixiang* sacrificial activity of the Zhang family in Wulajie Town, Jilin Province). Photo: Zheng De, 2014.

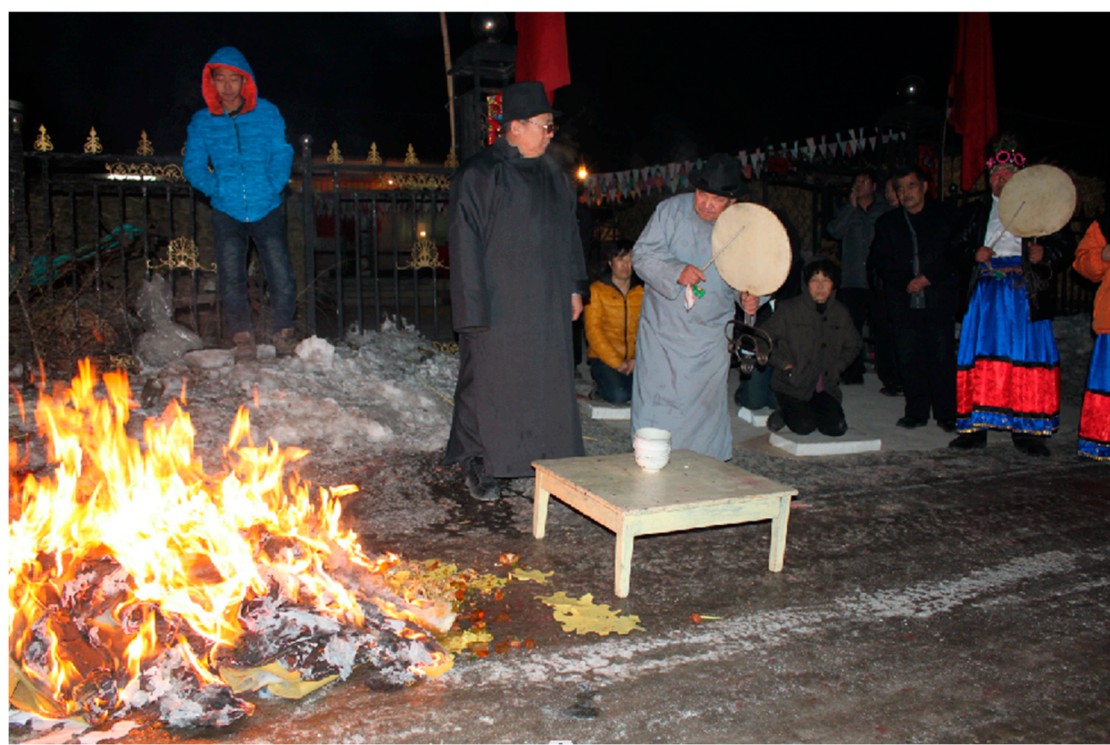

**Figure A6.** The banner effigies were burned outside the courtyard, and the family knelt to send the gods and ghosts away (18 February 2014, *Qixiang* sacrificial activity of the Zhang family in Wulajie Town, Jilin Province). Photo: Zheng De, 2014.

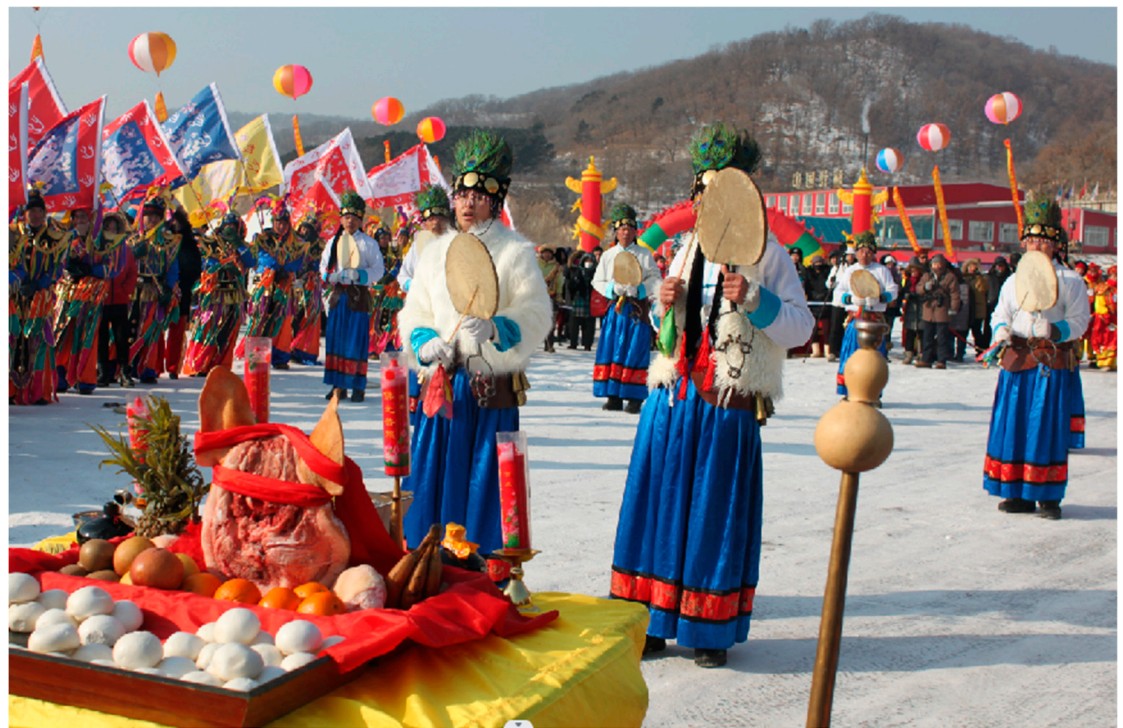

**Figure A7.** Drums were beaten and divine songs were sung to worship the gods. The banner effigies standing at the table have a gourd-shaped top, which is eye-catching (On 17 January 2016, Zhao Hongge (赵洪阁), Taiping Township, Yongji County, Jilin Province, led the incense squad to hold a wake-up ceremony at Zhuanshan Lake, Yehe City). Photo: Zheng De, 2016.

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
