# Peer review of "Re-Exploring Origins of the Qixiang Sacrificial Rite Practiced by the Han Army Eight Banners in Northeast China"

_religions, doi:10.3390/rel14020195_

Round 1

Author Response

Response to Reviewer 1 Comments

Point 1: Line 41-46. The Eastern Expedition theory was firstly proposed by Lu Xun (Please see Lu. 1988: 229-247. 程迅《满族陈汉军烧香礼俗与唐王征东》,吉林省民族研究所编《萨满教文化研究》). Fu Yuguang is also the supporter of this opinion (Please see Fu. 2010: 175. 富玉光, 满族萨满文化遗存调查). As best known scholar in the Manchu
shamanism, he should be included. 

Response 1: Thank you very much for your review. This problem is solved in lines 29-36, which supplements the views of Cheng Xun and Fu Yuguang and relevant references.

Point 2: Line 388-393. The words “Back to Normal” seem unclear, suggesting to use other words instead. Also, in my knowledge, hanging Xiliu occurs at the very beginning of the ritual before the insane-burning at the first stage. If I am not wrong, it should be corrected.

Response 2: This problem is solved in lines 372-378. The modified content is: Warnings and taboos

Point 2:Line 590-191. At the beginning of the conclusion, the author emphasizes that “the Qixiang sacrificial rite was a legacy of, and evolved from, the ancient Maji ceremony.” I have no problem with this statement. However, as concluding part, this is not enough. If the author adds a few theoretical words to state a syncretic nature of the qixiang culture, that will make his /her argument much stronger.

Response 3: This problem is resolved on lines 545-575. I used Stephan Feuchhwang's "Empire Metaphor" theory to explain the imitation of the Qixiang to the Maji; Leopold's religious "Synchretism" theory is also used to explain the phenomenon of the integration of the Maji, shamanism and the burning of incense by the Han people in the Qixiang sacrifice.

Reviewer 2 Report

This article is definitely worth publishing, but needs substantial revision.

The strengths are as follows:

1.      The sources used are valuable and not commonly used in English-language writing. You do well to use them and apply them to interesting and unusual material.

2.    It is good to begin by saying precisely what the QIxiang rituals are, though I'd welcome a bit more interpretation even then.

3.      The methodology in general is very good and well explained.

4.      The material is definitely new and interesting, and you have definitely made a contribution to knowledge in this field.

5.   It is good to include some pictures you took during your field research. This adds to the interest and scholarly value of the article.

There are also some weaknesses.

1.      The balance between description and interpretation is far too heavily weighted towards description. I think more explanation of what the whole things means, and of the political/cultural significance of these rituals would greatly improve the article. Even some mention of how these rituals are part of the history of China’s performing arts would be interesting, though not essential.

2.      We are told this article is designed as part of a special issue on “The Revitalization of Shamanism in Contemporary China”. Of course, that is fine. However, I think it would be good to explain why it is relevant to that overall topic. There is not much direct reference to shamanism in the article, let alone to shamanism in contemporary China. I think it can be readily be explained as relevant to shamanism in contemporary China, but it does need more material specifically dedicated to doing that.

3.      A bit more analysis of the sources would be useful. The primary Chinese sources are, of course, the essence of what you are using. But it would be interesting to know of material in any other language, such as Manchu or English.

4.      A bit more explanation of the theoretical basis of the article would be useful. It is too descriptive and not analytical enough.

5.      I am a bit confused by the use of Chinese characters. Why are they so regularly used for the titles of books, but never of the authors or other people? This is especially obvious in the list of references, where characters are always used for the titles of books, but not of the authors.

6.      The article is not well written and desperately needs editing. There are quite a few grammatical errors. Here is just one of numerous examples: on lines 27-28, the subject of the sentence is actually “Research”, which is singular, so the succeeding verb should be “has”, not “have”.

But, more generally, the language seems to me unnecessarily dense and complicated, which means that the main points do not come through clearly. Two examples can illustrate this.

One is that the paragraphs are too long and dense.

In explaining the five theories (lines 35 to 77), it would definitely be best to give a separate paragraph to each one. Otherwise it is too dense and difficult to follow, especially since the material is very unfamiliar. Another example is the explanation of the three research methods. Each method should have its own paragraph, not crush them all into one. It gives the article a feeling of denseness and complexity, which is unnecessary.

Some sentences are also too long. For example, in the paragraph refuting the theories (lines 66-77), it would be better to use a whole sentence for each theory, rather than putting them all in a long sentence, with separations made only by colons, not by full stops.

The abstract is dense and does not contribute as much to understanding the article as it should. It seems to me unnecessary to list the five theories of the origins and cultural significance of Qixiang. What we need is a clear and brief explanation of what the article is about, including the nub of the argument, the methodology, and why the article matters.

Author Response

Point 1: The balance between description and interpretation is far too heavily weighted towards description. I think more explanation of what the whole things means, and of the political/cultural significance of these rituals would greatly improve the article. Even some mention of how these rituals are part of the history of China’s performing arts would be interesting, though not essential. 

Response 1: Thank you very much for your review. In the revised article, I added the content of the political/cultural significance of the ceremony, and thought about the motivation and mechanism of the Han bannermen "imitating" the Maji ceremony and integrating the Shamanism and the Han people burning incense and offering sacrifices. These contents are reflected in lines 68-80, 447-458 and 557-588.

Point 2: We are told this article is designed as part of a special issue on “The Revitalization of Shamanism in Contemporary China”. Of course, that is fine. However, I think it would be good to explain why it is relevant to that overall topic. There is not much direct reference to shamanism in the article, let alone to shamanism in contemporary China. I think it can be readily be explained as relevant to shamanism in contemporary China, but it does need more material specifically dedicated to doing that.

Response 2: The amendments to this question are reflected in lines 68-74 and 576-588. This paper believes that Qixiang sacrifice is a unique part of Shamanism culture. Its evolution form and current recovery situation show a possibility of the development of Shamanism in China, that is, the integration of multiple religions produces a new form.

Point 3: A bit more analysis of the sources would be useful. The primary Chinese sources are, of course, the essence of what you are using. But it would be interesting to know of material in any other language, such as Manchu or English.

Response 3:At present, the research on Qixiang culture is mainly carried out in the field of Chinese, so the materials used are mainly from Chinese. However, this paper supplements theoretical materials from English sources, including the theories of Feng Qu, Leopold, Anita M., Jensen, Jeppe Sinding, Erdeni, Gagai, Kuerlen, Stephan Feuchhwang and other scholars.

Point 4: A bit more explanation of the theoretical basis of the article would be useful. It is too descriptive and not analytical enough.

Response 4:The revised article uses Stephan Feuchhwang's "Empire Metaphor" theory and Leopold's  "Synchretism" theory.

I used Stephan Feuchhwang's "Empire Metaphor" theory to explain the imitation of the Qixiang to the Maji;

Leopold's religious "Synchretism" theory is also used to explain the phenomenon of the integration of the Maji, shamanism and the burning of incense by the Han people in the Qixiang sacrifice.

Point 5:  I am a bit confused by the use of Chinese characters. Why are they so regularly used for the titles of books, but never of the authors or other people? This is especially obvious in the list of references, where characters are always used for the titles of books, but not of the authors.

Response 5:Thank you for your review. The author of the work has been indicated in the revised references. There are several reasons why only the title of ancient Chinese literature is marked without the author:

First, ancient documents are of a long time, and the specific author's name cannot be confirmed.

Second, many ancient documents were jointly completed by many people at different times, and some of these people had their names recorded, while others had no names recorded.

Third, the title of ancient Chinese documents is unique, and the corresponding documents can be found according to the title.

Fourth, many ancient documents were unearthed through archaeology.

Point 6: The article is not well written and desperately needs editing. There are quite a few grammatical errors. Here is just one of numerous examples: on lines 27-28, the subject of the sentence is actually “Research”, which is singular, so the succeeding verb should be “has”, not “have”.

But, more generally, the language seems to me unnecessarily dense and complicated, which means that the main points do not come through clearly. Two examples can illustrate this.

One is that the paragraphs are too long and dense.

In explaining the five theories (lines 35 to 77), it would definitely be best to give a separate paragraph to each one. Otherwise it is too dense and difficult to follow, especially since the material is very unfamiliar. Another example is the explanation of the three research methods. Each method should have its own paragraph, not crush them all into one. It gives the article a feeling of denseness and complexity, which is unnecessary.

Some sentences are also too long. For example, in the paragraph refuting the theories (lines 66-77), it would be better to use a whole sentence for each theory, rather than putting them all in a long sentence, with separations made only by colons, not by full stops.

The abstract is dense and does not contribute as much to understanding the article as it should. It seems to me unnecessary to list the five theories of the origins and cultural significance of Qixiang. What we need is a clear and brief explanation of what the article is about, including the nub of the argument, the methodology, and why the article matters.

Response 6:Thank you very much for your review.

In the article, I modified the syntax error, including the part you pointed out.

I introduce the five theories of Qixiang in five paragraphs (lines 25-56), and the research methods of this article in three paragraphs (lines 108-132), which can make the structure of the article clear.

The revised article "Abstract" deleted unnecessary introductory words about the five theories, supplemented the focus of scholars' debate, the understanding of the nature of Qixiang and the significance of Qixiang's research.

Point 7: It is good to include some pictures you took during your field research. This adds to the interest and scholarly value of the article.

Response 7: In the last part of this article, there is an appendix, which is the photos I took in the field survey, which helps to explain the content of the article.

Round 2

Reviewer 2 Report

This version is much better than the earlier one. It copes with the criticisms raised very well. It more just about ready for publication now.

It still needs editing and some minor changes need to be made.

Here are five examples.

On line 76, for "a grand and full of topics" read "a large one with many important topics"

On 78, Army is twice misspelt as Armey

On line 661 Cheng xun should read Cheng Xun

Line 683, Stephan Feuchtwang's surname is Feuchtwang. His name should be put under F in the reference list, not S.